# Long-term effect of middle ear disease on temporal processing and P300 in two different populations of children

**Leticia Reis Borges** [1]*, **Milaine Dominici Sanfins**[1], **Caroline Donadon**[1], **Dani Tomlin**[2],
**Maria Francisca Colella-Santos**[1]

**1** Department of Pediatrics, State University of Campinas, Campinas, São Paulo, Brazil, **2** Department of
Audiology and Speech Pathology, the University of Melbourne, Melbourne, Australia

* leticiarborges@yahoo.com.br

doi.org/10.1371/journal.pone.0232839

of Medicine, PORTUGAL

**Data Availability Statement:** All relevant data are
within the paper and its Supporting Information
files.

## Abstract

### Background/objective

The effects of otitis media on the function of the central auditory nervous system in different
populations is unknown. Understanding how the history of otitis media affects children from
different nations will guide health professionals worldwide on the importance of adequate
auditory stimulus in childhood. For this reason, the aim of the present study was to investi-
gate the long-term auditory effects of middle ear disease on temporal processing and P300
in two different populations of children: Australian and Brazilian.

### Methods

Temporal processing tests (Frequency Pattern Tests–FPT and Gaps in noise—GIN) and
P300 were measured in 68 Brazilian and Australian children, aged between 8 to 14 years.
The Brazilian otitis media group (BrOM) and Australian otitis media group (AusOM) con-
sisted of 20 children each who had a documented history of otitis media. Control groups of
14 children (BrControl and AusControl) were also recruited from each country, all with no
documented history of otitis media.

### Results

The BrOM group showed significantly poorer performance (p<0.001) for FPT and the GIN
compared to BrControl. The P300 response showed significantly longer mean latencies (p =
0.02) compared to BrControls. The AusOM group also showed significant delayed latency
of P300 (p = 0.04) compared to the AusControl. The FPT showed significantly poorer perfor-
mance (p = 0.04) compared to AusControls. The two otitis media groups showed no signifi-
cant differences between each other on P300. Significant differences were seen however in
temporal processing tests performance between the two cohorts for the otitis media groups.
The BrOM group had significantly poorer responses (p<0.001) for FPT and GIN compared
to the AusOM group.

**Funding:** Yes Fundação de Amparo a Pesquisa do Estado de São Paulo grant number: 2013/15672-4.

**Competing interests:** The author have declared that no competing interests exist.

## Conclusions

These findings support that although differences exist between BrOM and AusOM groups, otitis media can be demonstrated to affect the underlying mechanisms of the P300 measures and behavioral auditory responses in two different populations of children.

## Introduction

Adequate sensory experience is essential for the developing central auditory nervous system. Reduced auditory input, early in life, may affect listening abilities later in life [1]. Otitis Media with effusion (OME) is a common disease, which produces a persistent or fluctuant mild conductive hearing loss that is characterized by fluid in the middle ear with no symptoms of acute infection such as fever or otalgia [2]. Studies indicate elevated hearing level thresholds between 10 and 20 kHz in children with a history of recurrent otitis media (ROM). Frequently, the tympanic membrane position is retracted with decreased mobility, opaque appearance and abnormal color [3, 4].

The high prevalence of the OM in early childhood decreases with age. Children younger than seven years old have an increased risk of OM due to their immaturity of the immune system and the function of the Eustachian tube, which ventilates the middle ear space and equalizes pressure with the external environment [5]. Several epidemiologic studies of OM in the EUA and Scandinavian countries have shown that this disease has a high incidence in infants and young children [6–8]. A group of researchers in Boston showed that 70% of all children had had at least one episode of acute otitis media and one third had experienced three or more episodes by three years of age [9]. More than 50% of Swedish children also have reported to suffer at least one episode of OM before the age of four, and about 5% suffer frequent recurrences. The highest incidence occurring between six and 24 months [10].

The most common treatment used for middle ear infection are antibiotics and tympanostomy with tube placement insertion [11, 12]. These treatments are able to normalize the function of the middle ear and allow improvements in hearing levels. However, children with ROM can show deficits in binaural hearing and auditory abilities even years after the OM has gone and pure-tone thresholds have returned to normal [13, 14]. In addition, longitudinal studies of auditory processing in children with prior OM indicate that the risk of CAPD increased with longer duration of the conductive hearing loss [15,16].

Zumach et al. [17] have described the importance of central auditory processing (CAP) assessment, especially in children with a history of OM. The temporal processing is important for the listener to be able to understand speech in quiet and background noise, as speech stimuli and other background sounds vary time [18]. Some children with auditory difficulties, as a result of OM, can have difficulty in understanding the auditory stimuli over time which may hinder the acquisition of speech, language and reading. For example, studies have shown a correlation of the middle ear disease in children and temporal processing disorder [19,20].

The influence of a history of OM in childhood on auditory evoked potentials (AEP) can be assessed by varied electrophysiological measures. Where the auditory brainstem response (ABR) has been used to examine synaptic events related to more peripheral sensory function, long latency evoked auditory potential (LLEAP) have been used to determine how physical acoustic energy translates into patterns of brain activity and contributes to perception in normal and hearing-impaired listeners.

The LLEAP reflects the neuroelectric activity of the thalamus and auditory cortex areas, which is related to the functions of discrimination, integration and attention. P300 is an

endogenous LLEAP made up by a positive wave with post-stimulation latency of approximately 300ms indicative of activity in brain areas responsible for specific functions such as attention and memory [21].

The auditory deprivation, as a result of episodes of OM in childhood, compromises the normal development and maturation of the brainstem, mid-brain and cortical structures along the auditory pathway due to a lack of auditory experience [1]. Degraded auditory signals result in dys-synchronized activity at the auditory cortex, which is represented by small amplitudes on the LLEAP waves [15].

A number of animal studies also support that a lack of auditory experience can produce neural effects central to the cochlea [22,23]. Correlations have also been demonstrated between changes in LLEAP and a history of OM [24].

Therefore, the purpose of this study was to investigate how the long-term auditory deprivation caused by episodes of otitis media in childhood affects two distinct populations of children: Australian versus Brazilian and how is this effect on the temporal processing and P300.

## Materials and methods

### Ethics statement

This retrospective and cross-sectional study carries approval from the Research Ethics Committee of State University of Campinas, under number 682/2010 and Human Research Ethics Committee of the Royal Victorian Eye & Ear Hospital under number 13/1117H. A written informed consent was obtained from all children's parents.

Data collection was completed by the author; in Brazil at the Audiology Laboratory of the Center for Studies and Research in Rehabilitation and in Australia at the University of Melbourne Audiology Clinic.

### Participants

A total of 68 children and adolescents aged from 8 to 14 years old at the time of assessment, male and female, participated. 34 participants were right-handed Brazilian-Portuguese native speakers and 34 were right-handed Australian-English native speaker.

Inclusion criteria for control and study groups included:

- normal hearing and immitance at the time of assessment, defined as pure tone audiometry thresholds below 20 dBHL at 250 to 8000 Hz;

- The immitance values consistent with normal middle ear function were type A tympanograms, with peak compliance within 0.3 to 1.3 mmhos at -100 to +20dPa pressure with the presence of 1kHz ipsi and contralateral acoustic reflexes in both ears at the time of assessment [25];

- Both AusOM and BrOM had the additional inclusion criteria of a history of three episodes of OME with mild conductive hear loss in the first five years of life documented on the medical report;

Children with more than three episodes of OME, behavioural or neurological disorders and/or genetic syndromes (including those using psychoactive medications) or attending speech therapy were excluded from the sample.

The participants were divided into four groups:

- The Brazilian OM group (BrOM) consisted of 20 children (11 males and 9 females, mean age 11.2 years old) who had a documented history of three episodes of OME between 2002 and 2008

- Australian OM group (AusOM) consisted of 20 children (7 males and 13 females, mean age 10 years old) who had a documented history of three episodes of OME between 2002 and 2008

- Brazilian control (BrControl) consisted of 14 children (4 males and 16 females, mean age 10.5 years old) with no documented history of otitis media by parents

- Australian Control (AusControl) consisted of 14 children (7 males and 13 females, mean age 11.5 years old) with no documented history of otitis media by parents

## Procedures and measures

All subjects completed the auditory temporal processing tests and P300. These behavioral tests were chosen because they have the advantage of being non-language based. The assessment was performed in one 90 minute session in a soundproof condition. All groups were tested by Affinity (Interacoustic) audiometer and TDH 39P headphones were used to deliver the temporal processing task stimuli. The P300 was collected using the Navpro © Biologic system. The audiometric devices were calibrated in both countries to avoid any possibility of influence in the results.

The temporal processing tasks were composed of two tests: Gaps-in-Noise (GIN) and Frequency Pattern Test (FPT).

**Gaps-in-noise.** The GIN, developed by Musiek et al [26] consists of a series of 6-second segments of broad-band noise with 0 to 3 gaps embedded within each segment. The gaps vary in duration from 2-20ms. The gap-detection threshold is defined as the shortest gap duration correctly identified at least four out six times. The children were instructed to indicate each time they perceived a gap. The GIN test was presented monotically in the random order and measures temporal resolution ability.

**Frequency pattern test.** The FPT, developed by Musiek [27] is composed of three 150-msec tones and 200-msec intertone intervals. The tones in each triplet are combinations of two sinusoids, 880 Hz and 1122 Hz, which are designated as low frequency (L) and a high frequency (H), respectively. Thus, there are six possible combinations of the three-tone sequence (LLH, LHL, LHH, HLH, HLL, and HHL). The subjects were instructed that they would hear sets of three consecutive tones that varied in pitch. The task of the subject was verbalising the frequency patterns (e.g., high-low-high, low-low-high). The FPT was presented monotically in the random order and verifies the temporal ordering ability. A percentage correct score per ear is determined.

These tests were applied by the same examiner in both countries to harmonize the results.

**P300.** The LLEAP evaluation was performed using P300.

The potential P300 was recorded with the active electrode positioned on the vertex (Cz), the reference electrodes on the ipsilateral mastoid and the ground electrode at the Fz position, according to the 10–20 system [28]. The right and left ears were assessed separately. The equipment utilizes two channels with a band pass filter of 1–30 Hz. The eliciting stimulus was delivered monaurally through the insertion of earphones at 75 dB HL. The infrequent target stimulus was a 2 kHz tone burst presented randomly with a probability of 20% and the frequent stimulus (non-target) was a 1 kHz tone burst presented with 80% probability (oddball

paradigm). The stimulus rate was one stimulus per second, with a total of 300 sweeps. A 700 ms time window was used and the analysis was based on the numerical values of the latencies (ms) and amplitudes (µV). The P300 was identified as the positive deflection (250–500 ms) after the complex N1-P2-N2 [29,30]. The participants were instructed to count the infrequent target tone and the examiner verifies, in the end, children´s performance by asking them how many infrequent targets were counted. Two researchers analyzed the P300 traces to avoid any influence on the results.

## Statistical analyses

Ear side and gender effects were included as interactions. Groups, gender and side were fixed variables. The responses of the two ears were compared in each group using ANOVA. The significance level was determined to be 5%.

## Results

The analysis showed the effects of otitis media on FPT, GIN and P300 for each country, compared to their own control group, and then responses between populations.

There were no significant differences between the two ears for the Australian and Brazilian groups for FPT, GIN and P300. Therefore, data from two ears were combined for further statistical analysis.

### BrControl x BrOM

The statistical between groups showed significantly difference on temporal frequency pattern sequence test and GIN. As can be seen on and Table 1. The OM group had poorer performance compared to control group.

The P300 responses showed significant longer mean latencies (22.2 ms, p = 0.02), compared to BrControls (Fig 1). There were no significant differences between BrOM and BrControls for amplitude (p>0.05).

### AusControl x Aus OM

Table 2 shows the descriptive statistics between Australian groups for FPT. The AusOM group showed lower results for FPT.

Fig 2 reveals significant delayed latency on P300 (22 ms, p = 0.04) in the AusOM group compared to the AusControl. No statistical difference was found for amplitude between Australian groups.

### Brazilian versus Australian

The AusControl and BrControl groups showed no significant differences in behavioral responses and P300 measures (p>0.05).

Significant differences were seen in Table 3 between countries. The BrOM group had significantly poorer performance for FPT and GIN compared to the AusOM group.

**Table 1. Performance for FPT and GIN test based on Brazilian groups.**

| Tests | Ear | BrControl | | Ear | Br OM | | p-value | F value | Pr (>F) |
|---|---|---|---|---|---|---|---|---|---|
| | | Mean | SD | | Mean | SD | | | |
| FPT naming | 28* | 75.7% | 22.1 | 40* | 40.8% | 22.9 | <0.001 | 85.938 | 0.000 |
| GIN | 28* | 4.35 ms | 0.79 | 40* | 6.15 ms | 1.59 | <0.001 | 33.438 | 0 |

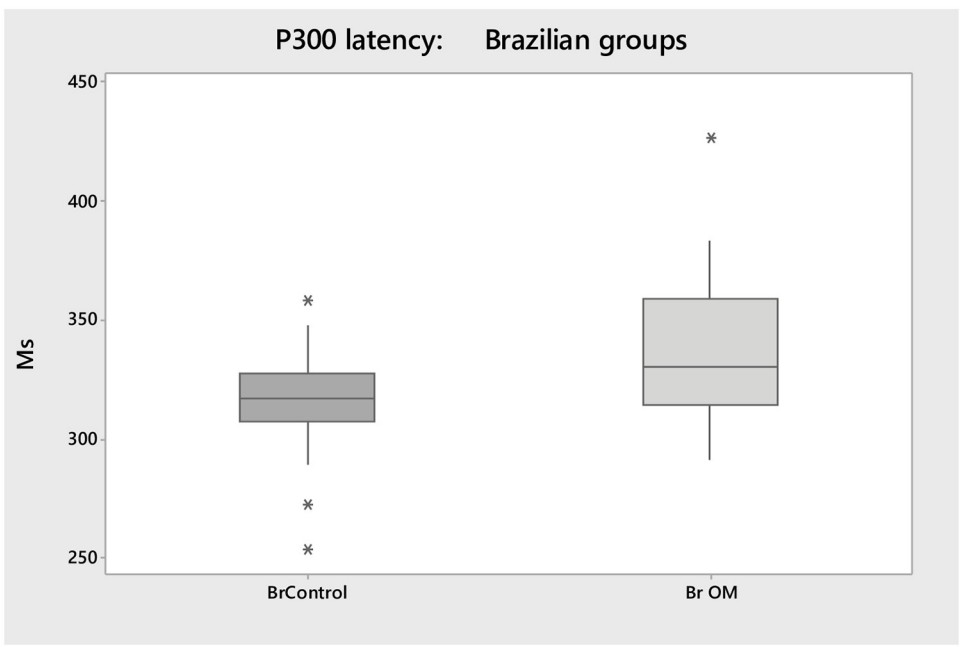

**Fig 1. Box plots shows the median, interquartile, and range of latency (ms) on P300 for both Brazilian groups.**

Table 4 shows the descriptive statistics between the two otitis media groups on P300 latency and amplitude measures. No significant differences were found between each other on P300 responses (p>0.05).

## Discussion

This study was undertaken to investigate the long-term auditory effects of middle ear disease on temporal processing tests and P300 in two different populations of children: Australian versus Brazilian. The children with reported history of OM from both countries demonstrated significantly longer latency on P300 measures and poorer performance on behavioral auditory responses. However, Brazilian OM group had significantly lower responses on FPT and increased responses on GIN compared to Australian children with the same disease.

### Effects of OM on temporal processing in children

Temporal processing can be defined as the perception of sound or the alteration of sound within a restricted or defined time domain. Temporal processing is divided into four different abilities: ordering, integration, masking and resolution and in this study, the abilities tested were ordering and resolution [31, 32]. Temporal ordering or sequencing refers to the processing of multiple auditory stimuli in their order of occurrence. This phenomenon has been

**Table 2. Performance for FPT and GIN based on Australian groups.**

| Tests | Ear | AusControl | | Ear | Aus OM | | p-value | F value | Pr (>F) |
|---|---|---|---|---|---|---|---|---|---|
| | | Mean | SD | | Mean | SD | | | |
| FPT naming | 28* | 87.6% | 12.5 | 40* | 74.5% | 24.2 | **0.040** | **7.5218** | **0.0079** |
| GIN | 28* | 4.42 ms | 0.53 | 40* | 4.52 ms | 0.64 | 0.64 | **0.275** | **0.602** |

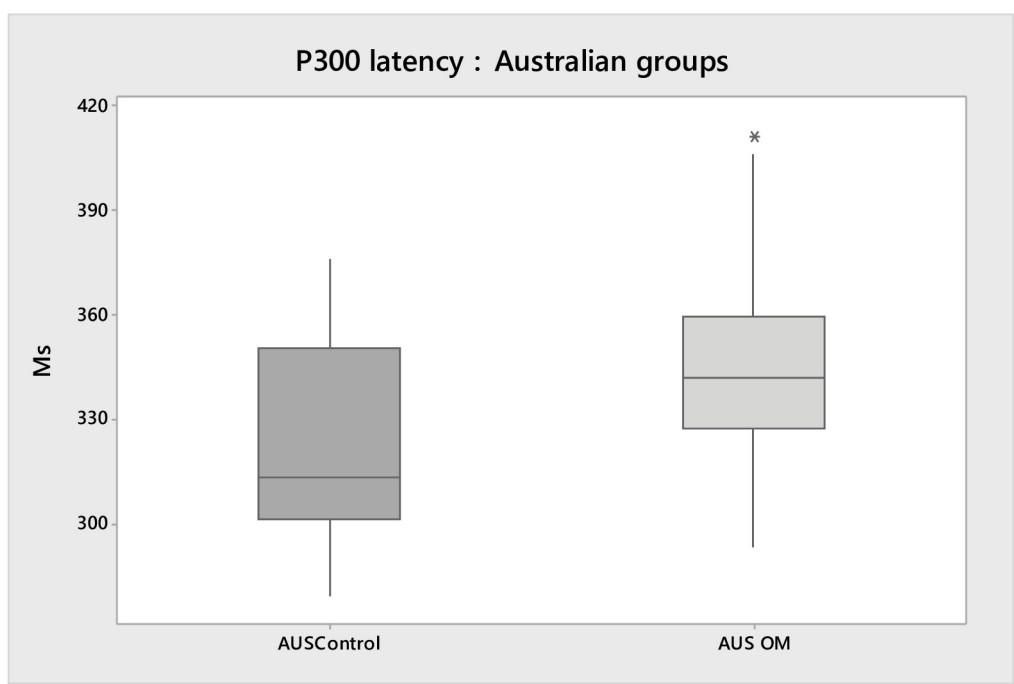

**Fig 2. Box plots shows the median, interquartile, and range of latency (ms) on P300 for both Australian groups.**

extensively investigated because of its importance in speech perception. Temporal discrimination or resolution refers to the shortest time in which a person can discriminate between two auditory signals [31].

Several studies have suggested that the temporal processing is a key factor to developing and understanding speech [31, 33, 34]. Previous research has also reported that children with a history of recurrent OME demonstrate speech and language deficits [35, 36]. The results of temporal processing tests in both Australian and Brazilian children showed that the temporal ordering ability was significantly lower in children with a history of OM compared to controls

**Table 3. Performance on temporal processing tests based on Brazilian and Australian OM groups.**

| Tests | Ear | BrOM | | Ear | AusOM | | p-value | F value | Pr (>F) |
|---|---|---|---|---|---|---|---|---|---|
| | | Mean | SD | | Mean | SD | | | |
| FPT naming | 40* | 40.8% | 22.9 | 40* | 74.5% | 24.2 | <0.001 | 41.381 | 0.000 |
| GIN | 40* | 6.15 ms | 1.59 | 40* | 4.52 ms | 0.67 | <0.001 | 32.598 | 0.000 |

**Table 4. P300 latency and amplitude results between Brazilian and Australian OM groups.**

| Wave | BrOM | | | AusOM | | | p-value | F value | Pr (>F) |
|---|---|---|---|---|---|---|---|---|---|
| P300 | Ear | Mean | SD | Ear | Mean | SD | | | |
| Latency (ms) | 40* | 338.6 ms | 33.7 | 40* | 345.3 ms | 31.5 | 0.36 | 0.308 | 0.581 |
| Amplitude (µV) | 40* | 5.6 | 1.34 | 40* | 6.25 | 2.58 | 0.090 | 0.906 | 0.344 |

(Tables 1 and 2). The temporal resolution ability also showed lower thresholds only in the Brazilian OM group. This is consistent with recent studies, which have demonstrated the effect of OM on auditory temporal processing in children [19, 20, 37]. Similarly, in animal studies, it was shown that 5–8 months of continuous bilateral ear plugging in ferrets resulted in a substantial aggravation of temporal resolution ability [38]. However, Hartley et al. [39] investigated the impact of otitis media with effusion (OME) in children and found no difference on temporal resolution ability in children with and without OME. It was noted that the cohort in the study did not have enough episodes of OME to cause auditory deprivation, which may explain why a temporal hearing deficit was not identified. Thus, reduced temporal abilities of children with a history of OM are suggested to be consequences of auditory deprivation as a result of repeated periods of hearing loss.

## Effects of OM on P300 in children

The LLAEP test is becoming an important and widespread clinical tool for the assessment of human cognition. The P300 waveform of children with suspected attention impairments can be compared against normative latency values to quantify the amount of dysfunction [40].

Children with a history of OM tend to have fluctuating hearing sensitivity during the first few years of life, which is the same period that auditory brainstem and cortical structures show the greatest development [41]. An early history of OM in children has been associated with a number of sensory, attention and social difficulties that are suggestive of changed brain function [42]. Previous studies have described P300 delays in children with learning or attention difficulties [40, 43]. There is some evidence to indicate that early onset OM and consequent reduced auditory input affects the latency at cortical potentials [1]. However, only one study was found which have investigated the effect of OM on P300. Shaffer [24] analyzed the responses of P300 in 36 children with minimal history of OM, children with a more significant history of OM and individuals with active OM. The author did not identify P300 responses in children and justified this finding due to the short time window used (500 ms). These findings are not consistent with the current study, which showed significant differences in P300 latency in both BrOM and AusOM groups (Figs 1 and 2) compared to controls. The association of OM on P300 may be explained by the fact that during the OM period the auditory cortex receives reduced auditory input from the brainstem, which, as consequence, effects the auditory processing at the cortical level. Yao & Sanes [44] telemetrically recorded from auditory cortex neurons in gerbils reared with developmental conductive hearing loss while they performed an auditory task in which rapid fluctuations in amplitude are detected. The authors compared the responses of auditory brainstem temporal processing from each animal. The study found that developmental HL decrease behavioural performance, but did not alter brainstem temporal processing. Therefore, the simultaneous assessment of neural and behavioural processing revealed that perceptual deficits were associated with a degraded cortical population code that could be explained by greater trial-to-trial response variability.

## Effects of OM between Australian and Brazilian children

The high prevalence of OM, difficulties in diagnosis and variations in management increases the likelihood of conductive hearing loss, and its potential impact on language, cognition and auditory abilities [45].

The results between Australia and Brazil controls showed no significant differences on behavioral responses or P300 latencies. These results demonstrated that children with no history of otitis media had the same responses, despite the difference in social and economic aspects between the two countries.

The P300 latency and amplitude measures showed no significant differences between Brazilian OM and Australian OM groups, which means that both otitis media group were equally affected. Thus, the auditory cortex was affected by the recurrent episodes of otitis media across populations.

A significant difference was observed on temporal processing tests between the two cohorts for the otitis media groups. The Brazilian children had significantly poorer performance on FPT and GIN compared to the Australian children (Table 3).

These results may be explained by the socio-economic levels between Brazil and Australia. There is evidence that low social-economic status increases the severity of OM [46].

The Human Development Index (HDI) is a composite statistic of life expectancy, education, and income per capita indicators. A country scores higher HDI when the life expectancy at birth is longer, the education period is longer, and the income per capita is higher. The latest report was launched in 2019 and showing Australia in sixth place and Brazil in 79th place [47]. Ruben [48] has found that 60% of children with OM and low social-economic level had hearing and learning difficulties.

The treatment of OM can differ worldwide. The use of antibiotics produces improvement in short term, but does not influence the course of the disease in a long-term.

The Brazilian and Australian OM group had the same number of episodes of OM. Despite the same inclusion criteria, a possible explanation for the poorer performance on temporal processing tests in Brazilian OM children is that the period to start clinical treatments with medication in Brazil could be longer due to the long waiting lists that public services usually have. Therefore, this can prolong the length of time of the fluid on the middle ear and reduce auditory input. Families of a lower social level may have diminished access to appropriate health services and immunization [49,50].

## Limitations and future research

The length of time of each period of OM was not collected. Therefore, it was not possible to analyze the correlation of duration of OM on temporal processing and P300. Despite of this limitation, our results corroborated to previous studies in the literature, which demonstrated that a history of otitis media affects behavioral and electrophysiological responses. Further studies may also include the length of time of each period of OM and the length of time to start the medical approaches.

## Conclusion

These findings support that the effects of otitis media on the underlying mechanisms on temporal ordering processing occurs across populations. However, a significantly greater impact in temporal tests is observed on the Brazilian cohort. This could suggest that different culture and/or time of medical approaches to treat otitis media may have implications on the severity of the auditory temporal deficit observed. In addition, P300 appears to be sensitive for the deprivation effects of otitis media once both OM groups had longer latencies compared with those children without middle ear disease.

Early identification and careful clinical follow-up of children with a history of OM is indicated for successful management of this disorder in order to contribute for the maturity of the central auditory nervous system.

## Supporting information

**S1 Data.**
(XLSX)

## Author Contributions

**Conceptualization:** Leticia Reis Borges, Maria Francisca Colella-Santos.

**Data curation:** Leticia Reis Borges.

**Formal analysis:** Leticia Reis Borges.

**Funding acquisition:** Leticia Reis Borges.

**Methodology:** Leticia Reis Borges, Milaine Dominici Sanfins, Caroline Donadon, Maria Francisca Colella-Santos.

**Supervision:** Maria Francisca Colella-Santos.

**Writing – original draft:** Leticia Reis Borges.

**Writing – review & editing:** Leticia Reis Borges, Dani Tomlin, Maria Francisca Colella-Santos.

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
