## [Editor Report · Decision Letter 0]

29 Jan 2020

PONE-D-19-32188

Long-term effect of middle ear disease on Temporal Processing and P300 in two different populations of children

PLOS ONE

Dear Ms Borges,

Thank you for submitting your manuscript to PLOS ONE. After careful consideration, we feel that it has merit but does not fully meet PLOS ONE’s publication criteria as it currently stands. Therefore, we invite you to submit a revised version of the manuscript that addresses the points raised during the review process.

We would appreciate receiving your revised manuscript by Mar 14 2020 11:59PM. To enhance the reproducibility of your results, we recommend that if applicable you deposit your laboratory protocols in protocols.io, where a protocol can be assigned its own identifier (DOI) such that it can be cited independently in the future. For instructions see: http://journals.plos.org/plosone/s/submission-guidelines#loc-laboratory-protocols

We look forward to receiving your revised manuscript.

Kind regards,

Jorge Spratley, MD, PhD

Academic Editor

PLOS ONE

Additional Editor Comments:

This manuscript presents the results of a multicenter study, comparing children form Brazil and Australia, in respect to the putative influence of otitis media-related hearing deprivation early in life and temporal auditory cortex function.

The topic is provocative and the study is original. The English of the manuscript lacks care and must be fully revised by an English native speaker for the sake of readability. Overall, the article improved very much when compared to the initial submission.

The following questions are still open and should be addressed by the authors:

1. Material and Methods L.124: Were the two cohorts, from Brazil and Australia, age-matched?

2. L.134: A definition of Recurrent Otitis Media must be clearly depicted. The text is not clear in separating infectious acute otitis media (AOM) and non-infectious otitis media with effusion (OME). AOM can cause inner disfunction through middle ear/inner ear interactions, via bacterial toxins. Could this occurrence influence the results?

3. L158: Were the audiologic tests carried out in similar conditions and with equal audiometric devices. Calibrations could have influenced the outcomes?

4. Sometimes the scores reported in this type of central processing exams are examiner-dependent. Could the authors clarify how they did to harmonize the results from both centers?

5. L.228/9 It is confusing: No significant differences in behavioural responses for P300 were found and this is followed by a p<0.05.

6. L237/8: the same as above (L.228/9) applies.

7. L.272: elevation means aggravation? (Seems a literal translation)

8. L.275/8: How is “severe” OME defined. Is it worse to have intermittent OME than persistent OME in respect to auditory deprivation?

9. L.294: found means finding?

10. L.300: temetrically?

11. L.317/8 correct grammar

12. L.332 correct grammar

13. L.337: queues mean waiting lists?

14. L. 342/2 correct grammar

15. Ref #7 Author name must be revised

16. Ref #19 is incomplete

17. Ref #38 the journal reference is correct?

18. Ref #43 seems an unpublished information

19. Ref #46 is the only one with et al. Was it borrowed from elsewhere?

20. Ref #47 the url is missing

2. Please provide additional details regarding participant consent.

In the ethics statement in the Methods and online submission information, please ensure that you have specified whether consent was written or verbal/oral. If consent was verbal/oral, please specify:

a) whether the ethics committee approved the verbal/oral consent procedure,

b) why written consent could not be obtained, and

c) how verbal/oral consent was recorded.

If your study included minors, please state whether you obtained consent from parents or guardians in these cases.

3. We noticed you have some minor occurrence(s) of overlapping text with the following previous publication(s), which needs to be addressed:

https://doi.org/10.1155/2019/8930904

https://doi.org/10.7554/eLife.33891

In your revision ensure you cite all your sources (including your own works), and quote or rephrase any duplicated text outside the Methods section. Further consideration is dependent on these concerns being addressed.
---

## [Author Response · Author response to Decision Letter 0]

19 Mar 2020

This answer also is attached as response to reviewers in attach files.

The manuscript was fully revised by an English native speaker and the changes were highlighted in yellow. 

Below is the answer for each comment in bold and the number of the pages of each change in the manuscript:

Editor Comments:

1. Material and Methods L.124: Were the two cohorts, from Brazil and Australia, age-matched?

Yes, the age matched. The mean age of each group was described in the lines 142, 145, 148 and 151. 

2. L.134: A definition of Recurrent Otitis Media must be clearly depicted. The text is not clear in separating infectious acute otitis media (AOM) and non-infectious otitis media with effusion (OME). AOM can cause inner disfunction through middle ear/inner ear interactions, via bacterial toxins. Could this occurrence influence the results?

A definition of OME was described in the Introduction: line 58. When we described children with Recorrent Otitis Media we wanted to highlight the number of episodes, but all these children had OME. In order to provide a better understanding we replaced ROM for OME. Line 135. 

3. L158: Were the audiologic tests carried out in similar conditions and with equal audiometric devices. Calibrations could have influenced the outcomes?

We were very concerned about the devices calibration in both countries. The tests were applied by the same type of audiometric devices and ear phones to avoid any possibility of influence in the results. This information was included in the lines 156, 157, 159 and 160.

4. Sometimes the scores reported in this type of central processing exams are examiner-dependent. Could the authors clarify how they did to harmonize the results from both centers?

In both centers the GIN test and FPT were applied by the same examiner to harmonize the results. This information was written in the lines 180 and 181. 

5. L.228/9 It is confusing: No significant differences in behavioural responses for P300 were found and this is followed by a p<0.05.

We wrongly written “for” and “p<0.05” instead of “and” and “p>0.05”. The correct sentence was rewritten in the line 233.

6. L237/8: the same as above (L.228/9) applies.

The symbol “<” was changed to “>” in the Line 242.

7. L.272: elevation means aggravation? (Seems a literal translation)

This change was made. Line 271.

8. L.275/8: How is “severe” OME defined. Is it worse to have intermittent OME than persistent OME in respect to auditory deprivation?

The term “severe” means how many episodes of OME the children had. In the article cited the author suggested that the cohort study did not have enough episodes to cause auditory deprivation and reduce temporal abilities. The sentence was rewritten in the lines 274 and 275. 

9. L.294: found means finding?

We wrongly written the word “found” instead of “finding”. The correct sentence was rewritten in the line 293.

10. L.300: temetrically?

The correct word is telemetrically. Is was rewritten in the line 299.

11. L.317/8 correct gramar

The gramar was corrected in the line 317. 

12. L.332 correct gramar

The gramar was corrected in the line 331.

13. L.337: queues mean waiting lists?

Yes. We replace the word queues for waiting lists in the line 336.

14. L. 342/2 correct gramar

The gramar was corrected. Lines 341 to 346.

15. Ref #7 Author name must be revised

The Author name was corrected. 

16. Ref #19 is incomplete

The reference 19 was corrected. 

17. Ref #38 the journal reference is correct?

Yes, it is corrected. Now ir Ref #39

18. Ref #43 seems an unpublished information

This reference was removed. 

19. Ref #46 is the only one with et al. Was it borrowed from elsewhere?

This article has more than 6 authors. So after the sixth author we wrote et al. 

20. Ref #47 the url is missing

This reference has been updated and rewritten. 

All journal requirements were made.

---

## [Editor Report · Decision Letter 1]

23 Apr 2020

Long-term effect of middle ear disease on Temporal Processing and P300 in two different populations of children

PONE-D-19-32188R1

Dear Dr. Borges,

We are pleased to inform you that your manuscript has been judged scientifically suitable for publication and will be formally accepted for publication once it complies with all outstanding technical requirements.

With kind regards,

Jorge Spratley, MD, PhD

Academic Editor

PLOS ONE

---

## [Editor Report · Acceptance letter]

29 Apr 2020

PONE-D-19-32188R1 

Long-term effect of middle ear disease on Temporal Processing and P300 in two different populations of children 

Dear Dr. Borges:

I am pleased to inform you that your manuscript has been deemed suitable for publication in PLOS ONE. Congratulations! Your manuscript is now with our production department. 

With kind regards,

on behalf of

Professor Jorge Spratley 

Academic Editor

PLOS ONE